# Harnessing Genetic Resistance in Maize and Integrated Rust Management Strategies to Combat Southern Corn Rust

**DOI:** 10.3390/jof11010041

**Published:** 2025-01-07

**Authors:** Jiaying Chang, Shizhi Wei, Yueyang Liu, Zhiquan Zhao, Jie Shi

**Affiliations:** 1Plant Protection Institute, Hebei Academy of Agriculture and Forestry Sciences, Key Laboratory of Integrated Pest Management on Crops in Northern Region of North China, Ministry of Agriculture and Rural Affairs, China, IPM Innovation Center of Hebei Province, International Science and Technology Joint Research Center on IPM of Hebei Province, Baoding 071000, China; changjiaying@163.com; 2Hebei Universe Agriculture Science and Technology Co., Ltd., Zhangjiakou 075100, China; weishizhinb@163.com (S.W.); liuyueyang@lpht.com.cn (Y.L.); 3Academic Affairs Office, Hebei Agricultural University, Baoding 071000, China; zhaoban@hebau.edu.cn

**Keywords:** southern corn rust, maize, *Puccinia polysora*, resistance genes, molecular interactions

## Abstract

Southern corn rust (SCR) caused by *Puccinia polysora* Underw. has recently emerged as a focal point of study because of its extensive distribution, significant damage, and high prevalence in maize growing areas such as the United States, Canada, and China. *P. polysora* is an obligate biotrophic fungal pathogen that cannot be cultured in vitro or genetically modified, thus complicating the study of the molecular bases of its pathogenicity. High temperatures and humid environmental conditions favor SCR development. In severe cases, SCR may inhibit photosynthesis and cause early desiccation of maize, a decrease in kernel weight, and yield loss. Consequently, an expedited and accurate detection approach for SCR is essential for plant protection and disease management. Significant progress has been made in elucidating the pathogenic mechanisms of *P. polysora*, identifying resistance genes and developing SCR-resistant cultivars. A detailed understanding of the molecular interactions between maize and *P. polysora* will facilitate the development of novel and effective approaches for controlling SCR. This review gives a concise overview of the biological characteristics and symptoms of SCR, its life cycle, the molecular basis of interactions between maize and *P. polysora*, the genetic resistance of maize to SCR, the network of maize resistance to *P. polysora* infection, SCR management, and future perspectives.

## 1. Introduction

Maize (*Zea mays.* L) is a very significant cultivated crop which is extensively grown as a primary source of food, fuel, animal feed, and raw ingredients for use in pharmaceutical industries [1]. Across various ecosystems, maize offers an ideal habitat for a wide range of pathogens and diseases, particularly those species that move or disseminate from the south to the north in response to climate change [2]. Therefore, it is imperative to secure sustainable maize production in order to provide food for the growing world population and attain global food security. Southern corn rust (SCR), caused by an obligate biotrophic fungal pathogen *Puccinia polysora* Underw., is prevalent in most maize-producing areas and may result in yield losses of up to 50% or above during outbreaks [3]. Historical records indicate that SCR was first documented in western Africa in 1949, is prevalent in tropical and subtropical areas [3], has been documented in China since 1972, and is now acknowledged as one of the most widespread diseases in maize farming, impacting over 100 million acres each year, causing substantial economic consequences [4,5,6,7,8,9,10].

The infective, proliferative, and reproductive functions of *P. polysora* are highly reliant on its living hosts, and its asexual stage is distinguished by the production of urediniospores and teliospores. *P. polysora* mostly survives the winter as urediniospores in tropical areas, serving as the main source for starting maize infections. Following the initial infection, the continuation of SCR depends on the continual production of urediniospores, which is aided by climatic conditions such as wind and precipitation [6,11]. Following late-season planting, symptoms first manifest on leaves and then spread throughout the whole plant, including the leaves, stalks, leaf sheaths, and husks. This may lead to leaf necrosis and the total loss of photosynthetic regions, ultimately resulting in the death of the plant [12,13,14,15]. Despite the existence of multiple physiological races of *P. polysora* [16,17,18,19,20], and the reported resistance to them [13,21], most commercial hybrids cultivated in China and the United States are considered to be susceptible to *P. polysora* [22,23]. Recently, SCR outbreaks have progressively expanded to high-latitude regions and have caused global catastrophic damage in maize-producing areas due to climate change [24,25]. Hence, the management of SCR is of utmost importance in ensuring optimal global maize production.

Maize resistance to SCR can either be race-specific or non-race-specific. Race-specific resistance confers full resistance to some pathogens, but not to others, and is determined by single resistance (*R*) genes, exhibiting a degree of heritability [26,27]. Non-race-specific resistance confers partial resistance that does not depend on particular pathogen avirulence genes, hence permitting infection but diminishing pathogen growth. Pathogens possess remarkable adaptability due to prolonged survival pressures and environmental stimuli, resulting in significant pathogen variations. Since the maize–*P. polysora* interaction is host- or race-specific, more novel virulent strains might rapidly evolve, resulting in various race-specific resistance genes included into maize cultivars ineffective against SCR development [28,29]. Consequently, it is essential to investigate novel maize SCR resistance genes to provide sustainable preventive and control strategies for this disease. Efficient management of SCR may be achieved by using *R* genes and developing maize lines that have *R* genes. This review provides a detailed overview of the molecular interactions between maize and *P. polysora*, including the biological characteristics and symptoms of SCR, its life cycle, the genetic resistance of maize against SCR, the network of resistance to SCR, its management, and future perspectives.

## 2. Discovery and Distribution of Southern Corn Rust

*Puccinia polysora* is a biotrophic fungal pathogen that was initially documented on eastern gammagrass (*Tripsacum dactyloides*) in 1886, described in 1897 in Alabama, and subsequently identified on a sugarcane-related species within the grass genus *Erianthus* [30,31]. The fungal pathogen was first precisely identified on maize in the United States by Cummins [32], and several herbarium specimens from other countries that were first classified as common maize rust, *Puccinia sorghi*, were subsequently shown to be *P*. *polysora* [32]. Cummins (1941) demonstrated that *P. polysora* was prevalent in Central and South America prior to 1891 [32], as well as in Massachusetts, USA in 1879 [33]. Later, SCR caused great concern upon its unexpected emergence in West Africa in 1949 and 1950, leading to severe epidemics that rapidly proliferated, resulting in substantial yield losses of up to 50% in Sierra Leone, Dahomey, Gold Coast, Ivory Coast, Liberia, and Nigeria [3,34]. SCR erupted aggressively in Kenya, Uganda, Nyasaland, and many distant areas throughout the Indian ocean region, including Mauritius in 1952 [31]. In 1953, SCR outbreaks were reported in the Philippines, where they caused 80–84% yield losses in susceptible cultivars [35]. It was also later reported in Canada in 1954 [31], and in 1957 it was initially documented in India, where it caused mild infection in maize [36]. The occurrence of *P. polysora* in India was reaffirmed in the Mysore area of Karnataka [36]. The disease was mostly observed in the lower Mississippi River Valley and was regarded as a minor maize disease in the USA until 1972, with many outbreaks documented across several states from 1972 to 1979 [37]. Thereafter, SCR swiftly spread to Australia [38,39], Thailand [40], Japan [41,42], and China [43], among other nations. In October 1999, moderate to severe infection of this rust was observed during the Post-Entry Quarantine Inspection (PEQI) of maize crops in Bangalore [44]. In China, SCR was first detected in 1972 in Hainan Province [43], and since 1998, it has advanced northwards, causing significant yield losses in Shandong, Hebei, Shanxi, Zhejiang, Henan, and Jiangsu provinces, among others [9,45,46,47]. The risk of SCR outbreaks is exacerbated by factors such as changes in farming practices, climate change, and extensive cultivation of susceptible cultivars [47].

SCR is now regarded as a global disease due to its dissemination to over 110 countries throughout Africa, Asia, the Americas and Australasia [9]. Furthermore, SCR was firstly reported in South Dakota in 2020 by South Dakota State University [48], and in North Dakota in 2021 [49]. These reports demonstrate that *P. polysora* is advancing northward and suggest that it may be progressively acclimatizing to milder conditions in North Dakota, or that the climate in the Midwest is becoming more conducive to SCR [49]. The occurrence of *P. polysora* in North Dakota and the absence of resistance in temperate corn inbreds [6] indicate that SCR is a risk to maize production in North Dakota. The detection of SCR in North Dakota will enhance future extension efforts on disease diagnostics and equip corn producers with management strategies should the disease become a recurrent issue.

## 3. Biological Characteristics and Symptoms

### 3.1. Pathogen Taxonomy and Host Range

*Puccinia polysora* belongs to the kingdom Fungi, phylum Basidiomycota, class Pucciniomycetes, order Pucciniales, family Pucciniaceae, genus *Puccinia*, and species *P. polysora*. *P. polysora*, as an obligate biotrophic fungus, infects, grows, and reproduces only on living maize plant tissues [32]. It also infects other host such as silver plume grass (*Saccharum apopecuroides*), a grass allied to sugarcane (*Erianthus divaricatus*), teosinte (*Euchlaena mexicana*), and four species of eastern gamagrass *Tripsacum*, that is, *T*. *dactyloides*, *T. lanceolatum*, *T*. *pilosum*, *T*. *latifolium*, and *T*. *laxum* [12,50]. In contrast to common rust (*P. sorghi*), which necessitates *Oxalis conorrhiza* as an alternative host for its sexual reproduction, the alternate host for *P*. *polysora* remains a mystery [51,52].

### 3.2. Physiological Races

The identification of *P. polysora* physiological races has been performed by utilizing maize differentials with distinct genetic sources [17,18,20,53]. Several physiological races of *P. polysora*, including EA.1 [53], EA.2 [16], EA.3 [19], and PP3-PP9 [17,18], identifiable by the reactions they trigger on various maize lines, have been reported. Using six maize cultivars in Brazil, Casela and Ferreira identified 16 virulence patterns in sixty single-pustule isolates [20]. Previous studies identified many distinct, significant, race-specific SCR-resistance genes in maize, including *Rpp1* (imparting resistance to *P. polysora* races EA.1 and EA.3), *Rpp2* (imparting resistance to races EA.1, EA.2, and EA.3), and *Rpp9* (imparting resistance to race PP.9) [19,54]. Despite the effective use of *Rpp9* for SCR management over the last 30 years, the resistance in *Rpp9*-harboring cultivars has diminished due to the significant genetic diversity of SCR races in the southern United States [5]. In tropical regions, the persistent inadequacy of research on race-specific resistance to SCR has been ascribed to the presence of numerous pathogen races in these places [20], and, consequently, it is essential to identify and categorize the physiological races and virulence patterns of *P. polysora*.

### 3.3. Signs and Symptoms

Common rust and southern corn rust can affect maize production. Concurrent infections of both rusts might manifest on the same plant, thereby complicating the process of diagnosis [11]. Therefore, maize producers must be able to differentiate between common rust (*Puccinia sorghi*) and southern corn rust (*P. polysora*) in maize to facilitate prompt and successful management choices [55]. SCR is a more virulent disease that may need fungicide treatment when present in the field, while common rust is less virulent and its occurrence in the field does not inherently need treatment [55,56,57]. *P. polysora* specifically targets all the aerial components of maize plants, such as leaves, sheaths, and stems. The extent and fatality of its damage are much greater than that of typical rust damage induced by *Puccinia sorghi* Schw. [21]. The symptom expression of these two corn fungal diseases is quite similar, leading to frequent misidentification.

SCR starts with little circular-to-oval lesions often surrounded by a bright green-to-yellow halo, while common rust starts with lesions on leaves that appear as specks, which then develop into tiny, tan patches [57]. Common rust lesions are round-to-elongated and may manifest in clusters, usually found on both the upper and lower surfaces of the leaves or leaf sheaths and are dispersed throughout the leaf surface. In contrast to common rust, SCR lesions manifest not only on leaf tissues but also on the stem, husk, and leaf sheath [48]. When SCR lesions develop, they penetrate the epidermis of the leaf surface, mostly occurring on the top leaf surface, and they include pale orange to cinnamon-red urediniospores that form pustules early in the season. SCR pustules are often smaller, orange-brown, more circular, and more densely clustered, while common rust pustules have a rich red hue and may be dispersed over the leaf (Figure 1A,B) [48,58]. In severe cases, the SCR pustules cane be present in both leaf sheaths and ear husks (Figure 1D) [56].

*Puccinia polysora*’s infective, proliferative, and reproductive functions highly depend on its living hosts, and its asexual stage is notable by the production of urediniospores and teliospores. The urediniospores of SCR are unicellular, yellowish-to-golden, and echinulate, with 4–5 equatorial holes; in contrast, the teliospores are bicellular, chestnut brown, and exhibit angular-to-ellipsoid or oblong shape, while common rust urediniospores are usually round (Figure 1C) [33,55]. Treatment of urediniospores from these two fungi with 75% hydrochloric acid or 95% sulfuric acid results in the protoplast of *P*. *polysora* contracting into a spherical shape, whereas the protoplast of *P*. *sorghi* contracts into multiple spherical forms, and this distinction can be utilized to differentiate between the two species [9,59]. In common rust, teliospores infect alternative hosts to complete its life cycle [57]. In SCR, teliospores are produced in the late season or are produced completely and are covered by the epidermis, which complicates the observation [58]. These teliospores are not significant for the dissemination and progression of the disease, and the alternative hosts remain unidentified [52]. The progression of SCR is dependent on urediniospores transported northward from tropical areas [57].

The fungal spores are transported by north-blowing winds from tropical regions to temperate maize fields, where they have the potential to infect any maize plant they come into contact with, provided those climatic circumstances are suitable. Common rust usually occurs earlier in the season since it thrives in a temperature range of 60° to 77° F and requires moisture for successful infection and proliferation on a maize plant [60]. However, SCR usually occurs later in the growing season, since it thrives in a temperature range of 77° to 82° F, can continue to develop above 82° F, and can also be active in July and August when temperatures are generally high [60]. SCR requires less moisture and can spread with very little moisture. Typically, six hours of dew is sufficient moisture to cause rust infection [60]. *P. polysora* mostly survives the winter as urediniospores in tropical areas, serving as the main source for starting maize infections.

## 4. *P. polysora* Life Cycle

The life cycle and reproductive mechanism of *P. polysora* remain unclear [9,11,12,61]. Although maize is the major host of *P. polysora*, its alternative host has yet to be identified. Urediniospores function as both the primary and secondary sources of infection in the minimal cycle of the disease [6]. *P. polysora* lacks both the aecidial and pycnidial phases [12]. The production of teliospores is infrequent or nonexistent [6]. Every effort aimed at stimulating the germination of teliospores has been fruitless [6,12]. Despite their potential importance in ensuring the prolonged survival of the pathogen, the role of teliospores in the life cycle remains a mystery [3,12]. The amazing success of the uredo stage and its seamless continuation may have led to the inhibition of the sexual phase in the life cycle [12]. In light of the insufficient data, Cammack tentatively categorized *P. polysora* as a microcyclic and autoecious hamiform [12,61].

In China, SCR may manifest in the majority of summer maize regions where the ambient temperature promotes disease progression [9]. *P. polysora* can overwinter and persist year-round, maintaining the uredo stage in southern regions of China where maize cultivation occurs continuously. Despite being mostly classified as a tropical disease, SCR is also prevalent in the southern United States and sporadically spread throughout the remaining grain belt. However, in Canada and the majority of the United States, it does not survive the winter and must undergo recolonization annually [25,62]. The fungal spores are transported by north-blowing winds from tropical regions to temperate maize fields, where they have the potential to infect any maize plant they come into contact with, provided those climatic circumstances are suitable.

## 5. Molecular Interactions Between *Puccinia polysora* and Maize

The key components of plant innate immunity are pattern-triggered immunity (PTI) and effector-triggered immunity (ETI) [63]. Phytopathogens have evolved a plethora of effector proteins to suppress PTI in plants [64]. Plants utilize disease-resistance proteins to recognize pathogen effectors and trigger ETI, thereby counteracting these pathogen-derived mechanisms [65]. Numerous *R* genes from a diverse array of plant species have been identified, characterized, and successfully cloned, and the largest category of *R* genes encode NBS-LRR proteins [66]. Recently, two SCR resistance genes, *RppC* and *RppK*, encoding NBS-LRR proteins were successfully cloned and their cognate avirulence genes *AvrRppC* and *AvrRppK* in *P. polysora* were also identified [28,29]. Analysis of *AvrRppC* allelic diversity by sequencing the genomic locus *P. polysora* isolates PP.CN1.0, PP.CN2.0, and PP.CN3.0 established six *AvrRppC* alleles [28]. Furthermore, it was found that *AvrRppC^E^* and *AvrRppC^ref^* alleles were harbored in PP.CN1.0; PP.CN2.0 harbors *AvrRppC^C^* and *AvrRppC^F^* alleles; and *AvrRppC^A^* and *AvrRppC^J^* were harbored in PP.CN3.0. *RppC*-dependent HR was induced by the transient co-expression of *RppC* with the three avirulent alleles *AvrRppC^C^*, *AvrRppC^E^*, and *AvrRppC^ref^* in *N. benthamiana* leaves, but the virulent alleles *AvrRppC^A^, AvrRppC^F^*, and *AvrRpp*^J^ did not [28]. The infiltration of avirulent proteins AvrRppC^C^, AvrRppC^E^, and AvrRppC^ref^ into *RppC*(+) maize leaves induced HR, but the virulent proteins did not. These results suggest that allelic variation in *RppC* significantly influences the effectiveness of the *RppC*-dependent resistance responses [28]. *AvrRppK* overexpression in maize increased SCR susceptibility in transgenic plants, demonstrating that it is a key *P. polysora* pathogenicity factor. By suppressing chitin-induced MAPK cascade activation and reactive oxygen species (ROS) accumulation, *AvrRppK* specifically inhibited maize PTI (Figure 2). It is unclear whether RppK directly or indirectly recognizes AvrRppK (Figure 2), although it may induce maize cells to produce a specific HR [29]. Comprehensive knowledge and continuous surveillance of the diversity and geographical distribution of the *AvrRppC* gene is essential to avert the evasion of effectors from *RppC*-mediated recognition and ensuing outbreaks of SCR [28]. The existence of virulent *AvrRppC* alleles indicates that extra genes (gene cassettes) should be utilized alongside *RppC* to enhance its longevity.

## 6. Genetic Resistance of Maize to Southern Corn Rust

Maize genetic resistance to SCR is crucial in global resistance breeding because it unravels innovative approaches for disease management. Genetic resistance against SCR can be race-specific resistance or non-race-specific resistance. Race-specific or qualitative resistance is controlled by single major-effect resistance genes, which may be either dominant or recessive, often providing race-specific, high-level resistance, but in a non-durable manner [68,69]. Single race-specific resistance genes impart substantial resistance to certain rust biotypes; however, easily inherited resistance may lead to the selection of virulent races. While qualitative resistance is more manageable in crop genetic research and breeding, partial resistance to pathogens may exhibit more durability than solely inherited resistance [68,69,70]. The durability of race-specific resistance genes in the field may be enhanced by the utilization of gene cassettes, strategic gene deployment, and multiline cultivars. Conversely, quantitative resistance usually has a polygenic basis and normally confers non-race-specific intermediate degrees of resistance [71]. However, the transmission of partial resistance has proven more challenging than that of hereditary resistance, owing to its assumed multigenic characteristics. Recombinants have been found that provide partial resistance, which seems to be non-race-specific and may be effective in the sustainable management of maize rusts [72].

The screening of resistant resources, together with the identification and mapping of resistance genes or quantitative trait loci (QTLs), forms the foundation for resistance breeding. Various studies have been conducted to elucidate the genetic and molecular mechanisms underlying maize resistance to SCR [7,24,28,29,73,74,75,76]. At least 20 SCR-resistance genes. including dominant resistance genes *Rpp1-Rpp11* [5,6,19], and significant resistance QTLs (*RppC* [77], *RppD* [78], *RppM* [7], *RppP25* [74], *RppQ* [4,22], *RppS* [23], *RppS313* [79], *RppCML496* [80] and *RppK* [29]) that confer resistance to SCR have been identified from a variety of maize germplasm resources to date (Table 1). Although most of these genes have been located on the short arm of maize chromosome 10, only three, *RppC* [28], *RppM* [7], and *RppK* [29], which encode typical NLR-type proteins, have been successfully cloned and characterized, thus unraveling essential genetic components for the development of SCR resistance via breeding [7,28,29].

The introgression of the *RppC* gene, along with its natural promoter, into susceptible maize inbred lines C01 and B104 resulted in significant resistance to SCR in the transgenic lines during field experiments [28]. Maize plants that were genetically modified to possess the *RppK* gene exhibited more robust resistance to all evaluated five isolates of *P. polysora*, compared to their wild-type progenitors. In contrast to *RppC*, the *RppK* allele, which confers resistance, was seldom in maize inbred lines and commercial hybrids [28,29]. Since both *RppC* and *RppK* are not found in the majority of maize germplasms, they are very valuable in the breeding of SCR resistance. Furthermore, combining *RppC* and *RppK* genes via breeding may augment SCR resistance [67]. Despite the typical fitness tradeoffs associated with the introgression of resistance genes into crops [81], no yield loss was detected in any of the evaluated maize hybrid lines in the absence of *P. polysora* after the introgression of the *RppK* gene [29]. Moreover, several significant agricultural features were unaltered by the introgression of the *RppK* gene. All findings corroborated that the *RppK* gene has significant potential in maize breeding.

Linked or association mapping has also identified QTLs that exhibit quantitative or partial resistance to SCR [4,6,75,76,82,83,84]. These QTLs provide supplementary genetic resources and crucial information for maize breeding [83,84,85]. Five QTLs designated as *qSCR3.04*, *qSCR5.07*, *qSCR6.01*, *qSCR9.03*, and *qSCR10.01* located on chromosomes 3, 5, 6, 9, and 10, respectively, were identified using recombinant inbred lines obtained from a cross between Ye478 and Qi319 [76]. *qSCR6.01* had the highest effective value, accounting for 24.15% of total phenotypic variation in two environments [76]. Furthermore, a significant QTL called *qSCR4.01* on chromosome 4 which accounted for 48–65% of the overall phenotypic variation was also identified from a recombinant inbred line obtained from a hybrid between CIMBL83 and Lx9801 [86]. Recently, through the integration of Bulk-Segregant RNA-Seq (BSR-Seq) and QTL mapping, Liu and colleagues have identified two novel QTLs, *qSCR4.05* and *qSCR4.08*, located on chromosome 4, that are associated with maize resistance to SCR [87]. The integration of recombinant inbred lines and chromosomal segment substitution lines populations can successfully dissect the SCR-resistance QTL research, which offers essential gene resource and genetic information for breeding resistant varieties [76].

**Table 1 jof-11-00041-t001:** A summary of maize identified and cloned SCR resistance genes.

Gene	Maize Variety	Chromosome	*R*-Gene Product	Reference
*Rpp1*	AFRO.29		Nd	[54]
*Rpp2*	AFRO.24 (SLP 20-4A)		Nd	[54]
*Rpp3-Rpp8*			Nd	[17]
*Rpp9*	PT186208		Nd	[88]
*Rpp10*	AFRO.761		Nd	[89]
*Rpp11*	AFRO.600		Nd	[89]
*RppP25*	P25	10S	Nd	[90]
*RppQ*	Qi319	10S	Nd	[22]
*RppD*	W2D	10S	Nd	[78]
*RppC*	CML470	10S	CC-NBS-LRR ptotein	[77]
*Rpp12*	Jiku12	10S	Nd	[91]
*RppS*	SCML205	10S	Nd	[23]
*RppS313*	S313 × PHW52	10S	Nd	[79]
*RppM*	Kangxiujing2416 (Jing2416k)	10S	CC-NBS-LRR ptotein	[8]
*RppCML496*	CML496	10S	Nd	[80]
*RppK*	K22 × DAN340	10S	CC-NBS-LRR ptotein	[29]
*RppSLN*	N531_R	10S	NBS-LRR ptotein	[92]

CC-NBS-LRR—coiled coil nucleotide binding site leucine-rich repeat, Nd—Not determined.

## 7. The Network of Maize Resistance to *Puccinia polysora* Infection

The maize disease resistance network can be explored by transcriptomics, proteomics and metabolomics, although there is limited knowledge about the proteomics and metabolic profiling during maize-*P. polysora* interactions. Integrated BSA-Seq, transcriptome, and physiological analyses identified many QTLs on chromosomes 1, 6, 8, and 10, as well as essential metabolic and defense pathways conferring SCR resistance [93]. Furthermore, a collection of 25 genes, including two CC-NBS-LRR genes, were established as candidate genes for a major-effect QTL on chromosome 10 [93]. One of the CC-NBS-LRR candidate genes was significantly upregulated in the resistant line L119A compared to the susceptible line irrespective of *P. polysora* inoculation. Furthermore, phenylpropanoid-derived lignin accumulation, particularly S lignin, was substantially increased in L119A following *P. polysora* inoculation, but remained unchanged in Lx9801, suggesting a key role of lignin in SCR resistance [93]. It was therefore proposed that *P. polysora* is likely to be recognized by immune receptors, such as the CC-NBS-LRR protein in L119A, but not in Lx9801. The immune receptors may subsequently activate essential transcription factors, which further modulate the expression of several defense -related genes, including pathogenesis-related genes, redox-related genes, and secondary metabolite biosynthesis genes [93].

Integrative multi-omics analyses identified four putative candidate genes associated with SCR response in maize. Included among these genes was *ZmHCT9*, which encodes the protein hydroxycinnamoyl transferase 9, and it was upregulated in susceptible inbred lines and associated with enhanced resistance to *P. polysora*. These findings offer significant insights into the genetic basis of maize SCR response and are beneficial for researchers seeking to identify possible genes associated with SCR resistance in maize [94]. Single-cell transcriptomics provides unparalleled insight into plant disease resistance mechanisms at the level of individual cell types, irrespective of histology or genetic markers [95,96,97]. Using high resolution single-cell RNA sequencing, various regulatory programs that enhance R99’s resilience across various leaf cell types were identified. This study revealed an immune-like condition in R99 leaves, marked by the expression of several fungi-induced genes without fungal infection, especially in guard and epidermal cells. It was also established that the fungi-induced glycoside hydrolase family 18 chitinase 7 protein (ZmChit7) confers resistance to SCR [98]. The differential proteins that were identified across cell clusters, such as calcium signaling, cysteine-rich receptor-like protein kinase, beta-glucosidase, chitinase, jasmonates, and redox regulation are directly associated with disease resistance. Comprehending the regulatory processes governing the expression of these genes is essential for elucidating the molecular basis of the interaction between maize and *P. polysora* [98]. Recently, two NBS-LRR genes (*Zmays10G000430* and *Zmays10G000440*) from a stable SCR-resistant introgression line N531_R were significantly upregulated upon *P. polysora* inoculation, demonstrating that they might be collectively involved in imparting resistance against SCR [92]. A proteomic analysis of SCR-resistant and -susceptible maize inbred lines found ZmREM1.3, a variably expressed remorin protein. Resistance to SCR was enhanced in *ZmREM1.3*-overexpressing plants, while reduced in a *ZmREM1.3* mutant, demonstrating the gene’s involvement in SCR resistance [24].

## 8. Transcriptional Regulation of Maize Resistance to *Puccinia polysora*

While mounting evidence has shown the transcriptional regulation of maize’s defensive response to abiotic and biotic stresses, the transcriptional regulation of maize resistance to *P. polysora* infection remains to be explored. There is currently limited knowledge about the molecular mechanisms and critical genes associated with resistance to SCR in maize. Furthermore, our comprehension of the alternative host of *P. polysora* and the functions of teliospores in host–pathogen interactions is insufficient. The ambiguous resistance route and prolonged conventional breeding techniques provide significant challenges in the advancement of better cultivars [99]. Therefore, there is an urgent need to clarify the molecular regulatory network that regulates maize–pathogen interactions, uncover defense or resistance-related genes and proteins, and develop molecular breeding approaches to improve disease resistance [98].

The plant defense response is governed by a complex regulatory network of immune receptors, transcription factors, and genes that regulate the accumulation of reactive oxygen species and biosynthesis of secondary metabolites, among other processes [100,101,102]. Upon detection of pathogen infection by immune receptors such as CC-NBS-LRR proteins, transcription factors, particularly from the WRKY and MYB families, play a role in the transcriptional regulation of defensive response genes [103,104,105]. Following the inoculation of *P. polysora*, the expression levels of 74 transcription factions, including 11 MYB and 17 WRKY transcription factors, were altered following *P. polysora* inoculation in highly resistant line L119A, indicating the activation of the transcriptional regulation network in response to pathogen infection [106]. The activation of upstream transcription factors resulted in the upregulation of downstream defense-associated genes in L119A, indicating a comprehensive activation of downstream defense genes (Figure 3). Twenty-two redox-related genes, including 11 genes encoding glutathione S-transferases and 11 genes encoding peroxidases, were increased in L119A upon inoculation, indicating that the redox state in L119A is advantageous for SCR resistance [93].

## 9. Economic Impact of SCR

The pathogen’s invasion and proliferation inside mesophyll cells may induce alterations in the leaf’s internal structure and result in reduced levels of chlorophyll, anthocyanin, and water content [106,107,108]. Through the reduction in photosynthetic area on the maize plant, SCR might deprive the plant of the necessary nutrients for grain fill, therefore diminishing the potential yield. Pustules have the potential to break the epidermis, therefore affecting the water regulation in the leaf. More severe infections may heighten susceptibility to stalk rot pathogens. During epidemic years, SCR may decrease maize productivity by as much as 50% or even 100% [109]. In recent years, the yearly maize yield loss attributed to SCR has significantly escalated due to rising winter minimum temperatures and the absence of SCR-resistant maize cultivars [109,110]. In China, the maize yield loss attributed to SCR in 2015 was estimated at 756 million kg, which was as high as 8.8 times the yearly average of 2008 to 2014 [109,110]. Furthermore, in 2015, the maize yield loss attributed to SCR was estimated at 3.193 billion kg in the United States and Canada, making it the sixth most severe disease for yield loss [62].

## 10. SCR Integrated Management

The efficacy of cultural methods in combating SCR is limited due to the dispersion of spores by wind from distant locations and their subsequent distribution across the region. Nevertheless, it is advisable to use resistant maize varieties when using late planting of maize, since late-planted maize may face an increased susceptibility to infection.

### 10.1. Deployment of Resistant Cultivars

The most economically efficient approach to control SCR in field corn is the development and deployment of resistant cultivars. The majority of maize products are susceptible to SCR, however there are a few resistance varieties. A variant of SCR resistance in maize is encoded by the *Rpp9* gene; however, in 2008, scientists in Georgia identified a novel strain of the SCR fungus that remained unaffected by the *Rpp9*-resistance gene [29]. In China, many maize cultivars, including Chengdan22, Zhengda619, Jundan20, and Nongda108, have been recognized as resistant or extremely resistant to SCR [111]. Another iteration of SCR resistance employs gene cassettes to generate resistance in the maize plant, therefore providing a degree of resistance against all identified strains of southern rust. Despite the presence of SCR pustules on maize with this novel genetic resistance, the severity of infections is reduced, and the effectiveness of fungicide management is enhanced.

### 10.2. Scouting

For generations, SCR management has always been a guessing game. Visual inspection is the conventional technique for identifying plant diseases in the field; nevertheless, it is ineffective and prone to error, hence it is of limited assistance in disease management [85,112]. Due to lack of resistance, often favorable environmental conditions, and the typical presence of the pathogen, SCR can manifest yearly. Growers must use a dual strategy that involves frequent scouting of the corn for the emergence of SCR and application of rust-managing fungicides under the appropriate development conditions. Weekly scouting is essential due to the rapid infection and dissemination, allowing for effective monitoring from the first stages of infection. [113,114].

### 10.3. Chemical Control

Foliar fungicides are very successful in controlling SCR, and a wide range of those products are specifically labeled for this purpose. The use of fungicides at or around the tasseling or silking growth stage, or VT/R1 and milk (R3) stage, offers the most significant advantage to optimize production potential. These applications are often implemented by aerial application, ground application, and chemigation via a center pivot [115,116]. Fungicides having diverse modes of action are often more efficacious in disease control and assist in managing fungicide resistance, however they may also incur higher costs [115]. Therefore, growers must be careful when using any fungicide to understand its mode of action. Repeated applications of fungicides with the same mode of action throughout a growing season may promote the emergence of fungal populations resistant to that mode of action or fungicide, complicating future management efforts [56].

While foliar fungicides are effective in safeguarding uninfected leaf tissue against southern rust, there is presently no defined treatment threshold for this disease [56]. Prior to applying a fungicide, carefully consider the presence or incidence of SCR in the region, crop growth stage, potential yield, specific irrigation method if relevant, and environmental factors conducive to disease development [56]. Yield losses are quantifiable when 5% of the leaf area is covered by SCR; hence, it is essential to protect the ear leaf against further SCR progression [117]. The total yield at dent (R5) growth stage has not been established; nevertheless, using a SCR fungicide past the onset of growth stage R5 will provide biological benefits but not economic advantages [56,117]. The effects of fungicides are not instantaneous; hence, depending on the infection cycle stage of the SCR pathogen, more rust progression may transpire before the fungicide achieves its maximum efficacy [117]. Despite the absence of a fungicide specifically targeting SCR, the application of 12.5% epoxiconazole SC at the maize booting stage significantly mitigated SCR damage by up to 85.54%, in contrast to the 20–40% effectiveness observed with conventional fungicides such triadimefon and zineb [118]. It was also found that the use of quinone outside inhibitor (Qol) or Qol + demethylation inhibitor fungicides during the tasseling stage was more effective in the presence of SCR and under environmental conditions conducive for rust growth [119].

### 10.4. Disease Prediction

To mitigate disease spread and minimize yield loss, spatial-explicit data on disease-infected plants is essential for precision crop protection, guiding pesticide application and other management practices [113,114,120]. The development of a rapid, non-destructive technology for detecting plant diseases such as SCR over large areas is a significant challenge [113,114]. Remote sensing technology has recently shown significant promise for the rapid and precise identification of plant diseases across numerous crops in a robust, non-destructive manner which minimizes pesticide use and guarantees food safety [113,114]. Disease management programs can be enhanced by the integration of disease prediction models and monitoring networks that serve as warning systems [109,121]. A molecular detection approach for monitoring SCR has been constructed [122], and the development of spectral disease indices, derived from in situ leaf reflectance spectra, has shown efficacy in identifying plant diseases in the field [85]. Furthermore, use of tools such as the Corn ipmPIPE (corn.ipmpipe.org/southerncornrust/) for tracking SCR observations throughout the growing season, is of great significance in making informed management decisions [115].

The incidence of SCR seems to be influenced by the local adaptation to climate change, wherein elevated temperatures and increased precipitation have beneficial impacts on sporulation, therefore creating ideal circumstances for the germination and growth of spores. Furthermore, the dissemination of spores is significantly influenced by wind spread, which has the potential to extend from the southern coast to the north, thereby exerting a wide-ranging regional influence [9,123]. Therefore, these climatic data are crucial for forecasting the seasonal dynamics of *P. polysora* spores and for projecting the distribution and danger of SCR development throughout different seasons. In a prior study conducted by Ramirez-Cabrald et al. [124], the CLIMEX model was used to predict the worldwide incidence of SCR in the future by using meteorological data, and they suggested a possible northward propagation of SCR by the end of the century, and by an elevated risk by 2050 [124].

Although CLIMEX and geographic information system (GIS) technology have been used as monitoring tools for predicting the potential distribution and probability of occurrence, they have limitations in predicting the severity of SCR, particularly due to their sudden and explosive nature in recent years [124,125,126,127,128]. Statistical models do not provide a comprehensive explanation of the biological mechanism statistics of the disease life cycle and are particularly limited in their ability to predict disease severity in northern epidemiological areas impacted by exotic infections. In order to tackle this difficulty, an ecoclimatic index risk prediction (EIRP) model for SCR, which is capable of replicating the biological processes that drive infection dynamics was recently developed [129]. Within a machine learning framework, this model ingeniously integrates meteorological and ecological physiological factors to provide predictions of complicated mechanism outcomes using extremely adaptable algorithms.

## 11. Limitations and Future Perspectives

SCR is unequivocally one of the most devastating diseases threatening global maize production. To combat this disease, breeders can use genetically resistant cultivars, which are both economical and devoid of chemicals. Despite being a catastrophic disease globally, the complete genome sequence and transcriptomic data for *P. polysora* remain unknown. The absence of effective genetic markers for *P. polysora* hinders studies on the disease’s migration. Conventional approaches for developing polymorphic simple sequence repeats (SSRs) markers are labor-intensive and costly, but transcriptome sequencing offers a high-throughput approach for examining SSRs [130,131]. Transcriptome sequencing is an efficient and economical technique for generating extensive transcriptomes and is utilized for annotating novel genes in both model and non-model species, as well as for developing molecular markers [132,133,134]. A recent study performed de novo assembly of the transcriptome and developed SSR markers for *P. polysora* [61]. This study unraveled new insights into the transcriptome, annotated unigenes, polymorphic SSRs, and population genetics of *P. polysora*. However, further research is required to analyze the population structure and functions of genes of *P. polysora*. Understanding the genetic diversity and variation in P. *polysora* may provide insights on the frequency and incidence of SCR, perhaps aiding in the development of more effective control strategies. Given its extremely invasive nature and potential for catastrophe, coordinated worldwide measures are essential to avoid the incidence and dissemination of pathogens.

Future breeding programs should focus on identifying both exploited and unexploited maize disease resistance genes, as well as their introgression, to produce many pathogen-resistant cultivars. Successful cloning of NLR proteins encoded by *RppC* and *RppK* will unravel novel insights into the molecular mechanisms underlying maize–*P. polysora* interactions. The high frequency of *RppC* among commercially prevalent SCR-resistant hybrids suggests its efficacy in conferring enough resistance to manage SCR epidemics without any negative impacts on yield. Moreover, this has enhanced the feasibility of monitoring the incidence of *AvrRppC* virulent alleles in the field to guide appropriate deployment strategies that increase the efficacy of *RppC* and perhaps prolong its durability. Numerous resistant germplasms extensively used in maize breeding programs have been generated without using marker-assisted selection for QTL resistance. Given the significant outbreak of SCR in recent years, it is imperative to generate highly efficient resistant germplasms to fulfill the requirements of germplasm enhancement.

## 12. Conclusions

SCR is a major global maize fungal disease, and its extensive prevalence and the consequent yield losses are causes of significant concern for all maize growers and researchers. Therefore, in this review, we discussed the biological characteristics of *P. polysora*, symptoms of SCR, the genetics of maize resistance against SCR and its management. A detailed understanding of maize defense mechanisms against SCR is essential for safeguarding the global food supply and for developing resilient, highly disease-resistant maize cultivars. Comprehensive collaboration is essential among researchers from various fields, including plant pathology, agronomy, climatology, epidemiology, computer science, and agro-meteorology, to enhance future investigations concerning the impact of climate change on the variable severity, prevalence, and distribution of SCR, as well as shifts in pathogen populations. This review will enhance a comprehensive grasp of the intricacies of SCR. It also provides a significant theoretical foundation for developing disease control measures; nevertheless, further research is required to understand the epidemiology, evolution, and dissemination of *P*. *polysora*.

## Figures and Tables

**Figure 1 jof-11-00041-f001:**
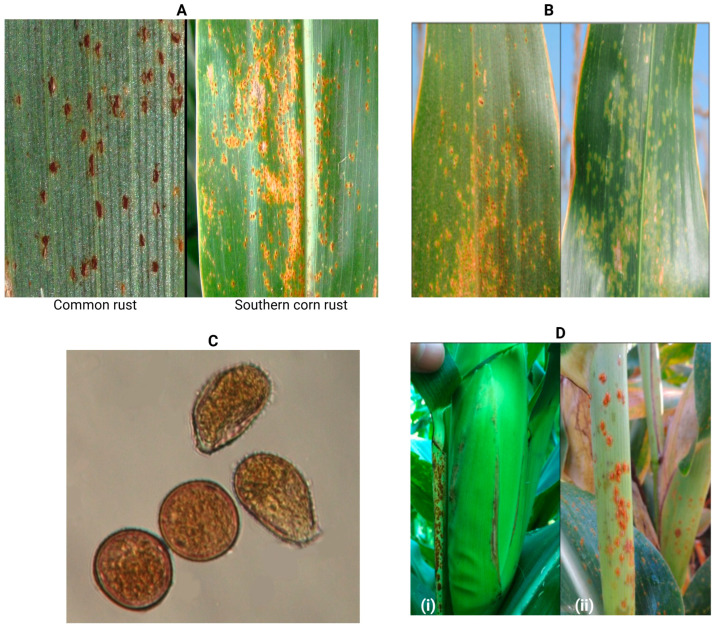
Symptoms of common rust and southern corn rust. (**A**). Magnified pustules of common rust fungus *P. sorghi* and southern rust fungus *P. polysora* (Photo courtesy of Tamara Jackson-Zeims) [55]. (**B**). Southern corn rust pustules occur primarily on the upper leaf surface (left) but on the lower leaf surface, only chlorosis is generally visible (right) (Photo courtesy of Bradley, C. et al. (2019) [56]. (**C**). Urediniospores (magnified 400×) of *P. sorghi* (common rust, lower left) and *P. polysora* (southern corn rust, upper right) (Photo courtesy of Tamara Jackson-Zeims) [55]. (**D**). In severe cases, southern corn rust pustules can be present on (**i**) ear husks and (**ii**) leaf sheaths (Photo courtesy of Bradley, C. et al. (2019) [56].

**Figure 2 jof-11-00041-f002:**
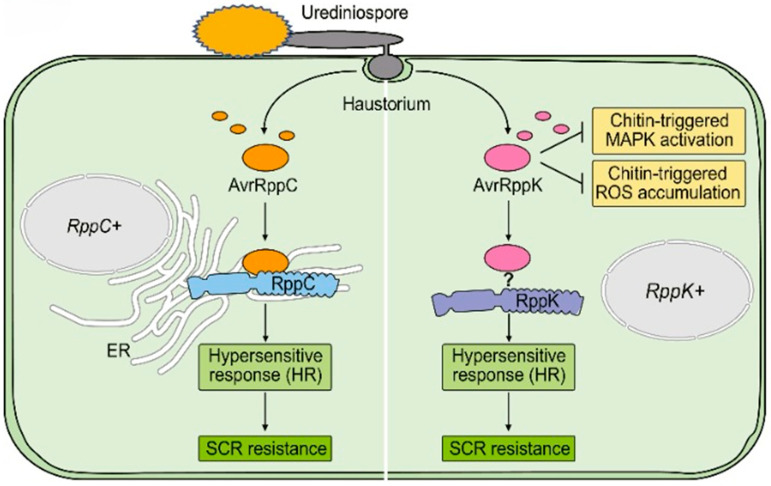
Maize resistance to southern corn rust mediated by *RppC* and *RppK*. Following the invasion of the maize cell wall, the haustorium of *Puccinia polysora* secretes AvrRppC and AvrRppK proteins into maize cells to promote pathogen colonization. The maize RppC protein recognizes *P. polysora* AvrRppC in the endoplasmic reticulum (ER). The interaction between RppC and AvrRppC triggers a hypersensitive response (HR) in maize cells and promotes host resistance to southern corn rust (SCR) (left). Secreted AvrRppK may impede chitin-induced MAPK activation and accumulation of reactive oxygen species (ROS), hence suppressing host pattern-triggered immunity (PTI). The maize RppK protein detects AvrRppK by an unspecified mechanism, triggers an HR in maize cells, and promotes host resistance to SCR (right) [67].

**Figure 3 jof-11-00041-f003:**
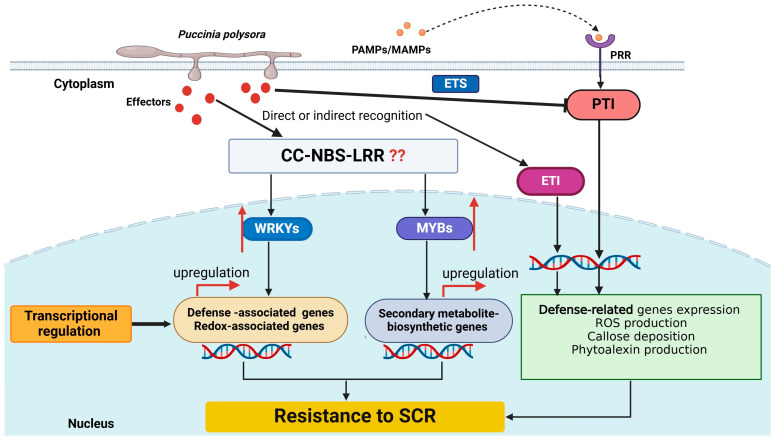
Schematic representation of transcriptional regulation of southern corn rust (SCR) resistance in maize. Detection of pathogen-derived conserved molecules (PAMPs/MAMPs) by pattern recognition receptors (PRRs), activates PAMP-triggered immunity (PTI). Pathogens induce susceptibility by interfering with the immune signaling network through the secretion of effectors, resulting in effector-triggered susceptibility (ETS). The direct or indirect recognition of pathogen effectors by immune receptors like CC-NBS-LRR proteins activates host defense responses to inhibit pathogen growth, and this is called effector-triggered immunity (ETI). Furthermore, pathogen infection perception activates WRKY and MYB transcription factors (upregulation indicated by red arrows), indicating the activation of transcription regulatory network. A series of secondary metabolite biosynthetic genes was upregulated along with the upregulation of MYB transcription factors, while a series of defense and redox-associated genes was upregulated along with the activation of WRKY transcription factors [93]. However, the specific CC-NBS-LRR proteins involved in the activation of the transcription factors and the molecular mechanism underlying the activation process are not yet known (indicated by question marks). ROS—Reactive oxygen species, CC-NBS-LRR—coiled coil nucleotide binding site leucine-rich repeat; red arrows indicate upregulation.

## Data Availability

Not Applicable.

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
