# Peer review of "Harnessing Genetic Resistance in Maize and Integrated Rust Management Strategies to Combat Southern Corn Rust"

_jof, 2025, doi:10.3390/jof11010041_

Round 1

Reviewer 1 Report

In this review, the interaction between Puccinia polysora, the causal agent Southern corn rust, (SCR) in maize plants was explored. The manuscript focuses on the molecular basis of genetic resistance to the disease, on the mechanisms of plant-pathogen interaction involved and on the disease management. This is a good review that summarizes important information associated with this Southern corn rust.

Line 77-P. polysora biological characteristics and SCR symptoms

Do not use abbreviations and acronyms in the title. See also below

Line 78 P. polysora

Use extended name at the beginning of the paragraph

Line 77-P. polysora biological characteristics and SCR symptoms

In this paragraph add some information about the taxonomic of P. polysora

Lines 144-157 ‘The key components of plant innate immunity are pattern-triggered immunity (PTI) 144 and effector-triggered immunity (ETI) [39]. Phytopathogens evolved a plethora of effec- 145 tor proteins to suppress PTI in plants [40]. Effectors serve as the primary virulence factors 146 of pathogens, whereas nucleotide-binding leucine-rich repeat (NLR) proteins constitute 147 the principal elements of the plant immune system, both of which are influenced by 148 evolutionary pressure [41], leading to significant diversity in effectors and NLR proteins 149 [39,42]. Plants utilize disease resistance proteins to recognize pathogen effectors and 150 trigger ETI, thereby counteracting these pathogen derived mechanisms [43]. Defense 151 responses facilitated by R genes that are activated upon pathogen effector recognition 152 by R proteins [44-46], often include hypersensitive response (HR), which induces rapid 153 programmed cell death at infection sites to curb invasive growth and proliferation of 154 pathogens in the tissues of the host plant [47-49]. To circumvent detection by host re- 155 sistance proteins, several core effectors exhibit polymorphism in various isolates. Further 156 research on NLR proteins and core effectors is necessary to elucidate why core effectors 157 are not eliminated in the arm-race [28]’

 This part can be reduced. The molecular mechanisms underlying plant-pathogen interaction are known to all pathologists and reported in numerous works.

Focus your review on P. polysora and maize interaction

Line 176: ROS accumulation,

specify ROS the first time

Lines 163-164

Add references

Lines 163-169

Add references

Lines 379- 393

Add some references

Author Response

Thank you for your suggestions. All the raised issues have been addressed in the revised version of the manuscript as below. Furthermore, the title of the manuscript has been changed according to Reviewer 2’s comments.

Point 1: Line 77-P. polysora biological characteristics and SCR symptoms
Do not use abbreviations and acronyms in the title. See also below
Response 
It has been revised – Line 121 

Point 2: Line 78 P. polysora
Use extended name at the beginning of the paragraph
Response 
It has been revised in all sections.

Point 3: Line 77-P. polysora biological characteristics and SCR symptoms
In this paragraph add some information about the taxonomic of P. polysora
 Response 
Some information about the Taxonomy of P. polysora has been added (Line 120 – Line 129)

Point 4: Lines 144-157 This part can be reduced. The molecular mechanisms underlying plant-pathogen interaction are known to all pathologists and reported in numerous works. Focus your review on P. polysora and maize interaction
Response 
It has been revised  (Line 230 – Line 234)

Point 5: Line 176: ROS accumulation,
specify ROS the first time
Response 
ROS has been defined  - Line 251 

Point 6: Lines 163-164
Add references
Response 
References have been added – Line 240, Line 245

Point 7: Lines 163-169
Add references
Response 
References have been added – Line 240, Line 245

Point 8: Lines 379- 393
Add some references
Response 
References have been added – Line 477, Line 479, Line 482, Line 484, Line 486, Line 489) 

Reviewer 2 Report

This peer-reviewed review is a detailed study in agricultural science and aims to describe a corn disease (southern corn rust) from the point of view of its causal agent - Puccinia polysora. The review is generally interesting and covers several aspects, in particular: some biological characteristics of P. polysora, the life cycle of this pathogen, molecular interactions between the pathogen and maize. Molecular genetic causes of maize resistance to the disease are considered separately. The following should be noted as shortcomings of the review:

1. Not enough illustrative material (there are only two figures and one table). I would like to see a Figure for section 7 "Transcriptional regulation of maize resistance to P. polysora", for example.  

2. The conclusions of the review need to be expanded. The conclusions need to discuss the significance of the review in more detail, and also clarify for whom and why it was written.

The review is generally good, only minor adjustments are needed:

1. Please, change a title of review. The title should cover the topic of the review more broadly and be more catchy.

Author Response

Thank you for your suggestions. All the raised issues have been addressed in the revised version of the manuscript as below.
This peer-reviewed review is a detailed study in agricultural science and aims to describe a corn disease (southern corn rust) from the point of view of its causal agent - Puccinia polysora. The review is generally interesting and covers several aspects, in particular: some biological characteristics of P. polysora, the life cycle of this pathogen, molecular interactions between the pathogen and maize. Molecular genetic causes of maize resistance to the disease are considered separately. The following should be noted as shortcomings of the review:

Point 1: Not enough illustrative material (there are only two figures and one table). I would like to see a Figure for section 7 "Transcriptional regulation of maize resistance to P. polysora", for example.  
Response
A figure has been added – Figure 3 – Line 404 – Line 422 

Point 2: The conclusions of the review need to be expanded. The conclusions need to discuss the significance of the review in more detail, and also clarify for whom and why it was written.
Response 
The conclusion has been modified – Line 573 – Line 586 

Point 3: The review is generally good, only minor adjustments are needed:
Please, change a title of review. The title should cover the topic of the review more broadly and be more catchy.
Response
The title has been changed to: Harnessing genetic resistance in maize and integrated rust management strategies to combat southern corn rust (Line 2 – Line 3)

Reviewer 3 Report

Great review article by Chang et al. I agree with the publication after revision which could improve the quality of the Manuscript. 

Great review article by Chang et al. Although main focus in this article is on the molecular interactions between pathogen and host, since this is review article, more information about the biology of pathogen are needed. For example I would like to read in section 2, what are the main differences between common rust of maize (P. sorghi) and southern corn rust (P. polyspora).

Some historical facts regarding the P. polyspora such as discovery, host range, pathogen reports worldwide in this section would also improve the manuscript.

What is the source for Figure 1? Is this original photo made by the authors or taken from Bradley et al (2019). Please be more specific with the Photo description (where and when the Photo was taken).

Author Response

Thank you for your suggestions. All the raised issues have been addressed in the revised version of the manuscript as below. Furthermore, the title of the manuscript has been changed according to Reviewer 2’s comments. 

Point 1: Great review article by Chang et al. Although main focus in this article is on the molecular interactions between pathogen and host, since this is review article, more information about the biology of pathogen are needed. For example, I would like to read in section 2, what are the main differences between common rust of maize (P. sorghi) and southern corn rust (P. polyspora).
Some historical facts regarding the P. polyspora such as discovery, host range, pathogen reports worldwide in this section would also improve the manuscript 
Response
More information about the biology of the pathogen has been added 
Differences between common rust of maize (P. sorghi) and southern corn rust (P. polyspora) have been added.
Discovery, host range and pathogen reports worldwide has been added. (Line 75 – Line 189)

Point 2: What is the source for Figure 1? Is this original photo made by the authors or taken from Bradley et al (2019). Please be more specific with the Photo description (where and when the Photo was taken).
Response 
This is not an original photo produced by the authors, but it was adopted from Bradley et al., (2019), and it has been indicated. The figure has also been modified. (Line 186 – Line 195)

Round 2

Reviewer 3 Report

The quality of the Manuscript has been improved after revision. I agree with publication. 

Accept